# HDX-MS: An Analytical Tool to Capture Protein Motion in Action

**DOI:** 10.3390/biomedicines8070224

**Published:** 2020-07-17

**Authors:** Dominic Narang, Cristina Lento, Derek J. Wilson

**Affiliations:** 1Department of Chemistry, York University, Toronto, ON M3J 1P3, Canada; dnarang@yorku.ca (D.N.); clento@yorku.ca (C.L.); 2Centre for Research of Biomolecular Interactions, York University, Toronto, ON M3J 1P3, Canada; 3Centre for Research in Mass Spectrometry, York University, Toronto, ON M3J 1P3, Canada

**Keywords:** hydrogen–deuterium exchange mass spectrometry, HDX-MS, protein dynamics, protein folding and misfolding

## Abstract

Virtually all protein functions in the cell, including pathogenic processes, require coordinated motion of atoms or domains, i.e., conformational dynamics. Understanding protein dynamics is therefore critical both for drug development and to learn about the underlying molecular causes of many diseases. Hydrogen–Deuterium Exchange Mass Spectrometry (HDX-MS) provides valuable information about protein dynamics, which is highly complementary to the static picture provided by conventional high-resolution structural tools (i.e., X-ray crystallography and structural NMR). The amount of protein required to carry out HDX-MS experiments is a fraction of the amount required by alternative biophysical techniques, which are also usually lower resolution. Use of HDX-MS is growing quickly both in industry and academia, and it has been successfully used in numerous drug and vaccine development efforts, with important roles in understanding allosteric effects and mapping binding sites.

## 1. Introduction

Detailed studies of protein structure and function are critical to understanding the mechanism of biological processes and the cause of various human diseases [1,2]. The vast majority of drugs available as therapeutics bind to proteins and modulate their structure-function relationship [3]. For small-molecule drugs, their impact on the target protein is typically the direct result of only a few non-covalent interactions, although changes in conformation or dynamics may be widespread through allosteric effects. In this case, challenges associated with achieving a molecular-level mechanism for protein function arise largely from the complexity of the target. For instance, various proteins that are an integral part of the cell signaling (and thus tempting drug targets) like NFκB, p53, or proteins associated with amyloid diseases like α-synuclein and tau proteins are abundant in intrinsically disordered regions (IDRs) and are not amenable to traditional structure determination techniques like X-ray crystallography or solution NMR. Adding to these challenges, an increasingly large fraction of biopharmaceutical products are protein/peptide-based drugs, with molecular weights that vary from few kilodaltons to megadaltons [4,5]. When proteins/peptides are used as drug molecules, the complexity of the drug/target interaction is magnified by conformational heterogeneity and the distribution of covalent modifications in the drug itself.

Hydrogen–Deuterium Exchange Mass Spectrometry (HDX-MS) is a powerful analytical technique that can provide useful information about solution phase protein conformation and dynamics, even for weakly structured or membrane-associated targets [6,7]. In conventional ‘forward’ HDX-MS experiments, proteins in ‘native-like’ H_2_O-based solvents are diluted into similarly buffered D_2_O solvents, causing labile hydrogens on the protein to undergo exchange solvent deuterium. The rate of this exchange is dependent on a number of controllable factors (i.e., pH and temperature), but also on protein structure and dynamics in the sense that hydrogens may be protected from exchange to varying degrees due to limited solvent access (i.e., tertiary structure) or hydrogen bonding (i.e., secondary and tertiary structure). The exchange rate is measured by collecting MS data at several intervals after exposure to D_2_O, resulting in ‘deuterium uptake vs. time’ profiles that reflect conformation and dynamics under particular conditions (e.g., in the presence or absence of a ligand). Deuterium uptake can be measured for the entire protein (global HDX) but is more often acquired using a ‘bottom-up’ workflow (see Section 2) that involves quenching of the exchange reaction followed by proteolytic digestion before LC/MS-based analysis. The advantage of this approach is that it provides a degree of ‘spatial resolution’ to the data, usually in segments of around five residues on average.

Unlike, X-ray crystallography and structural NMR, HDX-MS does not provide atomic resolution structures, although there are ongoing efforts to optimize the use of HDX data for parameterization of computationally predicted structures. At the moment, however, HDX-MS data are much more informative when viewed as a measure of conformational dynamics (or ligand-induced changes in structure/dynamics) superimposed onto a ‘reference’ high-resolution structure. The great advantage of HDX-MS is its versatility—virtually any system can be directly explored, including large biomolecular complexes, highly dynamic proteins and membrane-associated species. Ultimately, HDX-MS can be viewed as providing a ‘dynamic’ picture of protein function and pathogenesis that is complimentary to the largely ‘static’ high-resolution images provided by the classical tools of structural biology.

### Underlying Theory

Upon exposure to D_2_O, labile hydrogens bound to heteroatoms (N, O, or S) on proteins readily undergo an exchange with solvent deuterium. Of these, only hydrogens bonded to backbone amide nitrogens undergo this exchange on a timescale that is relevant to conventional HDX measurements, and these are also the sites where exchange is most sensitive to hydrogen bonding in secondary structures. Further discussion will therefore refer specifically to these sites. The overall exchange rate constant is highly dependent on the pH of the solvent as the process can be acid, base or water catalyzed. The ‘chemical’ HX exchange rate constant *k*_ch_, which assumes no contribution from protein structure, can be expressed by the equation:
*k*_ch_ = *k*_int,H_ [H^+^] + *k*_int,OH_ [OH^-^] + *k*_H2O_[H_2_O](1)
where *k*_int_,_H_, *k*_int,OH_, and *k*_H2O_ represent the intrinsic rate constants for the acid-, base-, and water-catalyzed reactions, respectively (Figure 1) [8,9,10]. In a base catalyzed reaction, a deuterated hydroxide ion attacks the amide proton and removes it. This leaves a negative charge on the amide nitrogen atom, which accepts the D^+^ ion from another D_2_O molecule and gets reprotonated (Figure 1a). Acid catalysis has two different competitive mechanisms. In one possible mechanism, a D_3_O^+^ molecule attacks the electronegative amide nitrogen atom and forms a transient complex from which proton is transferred to a water molecule (Figure 1b). In another possible mechanism, the protonation of an amide oxygen atom increases the acidity around the NH group, causing the removal of a H^+^ ion by water. The N atom is then protonated by another D_3_O^+^ molecule (Figure 1c). Studies on the dependence of the exchange rate of poly-DL-alanine on pH give a V-shaped curve with the lowest point around pH 2.54 (Figure 1d). Above this pH (and importantly at neutral pH where most HDX measurements are taken), the HX reaction is dominated by base catalysis. In a polypeptide, the intrinsic exchange rates (i.e., all *k*_int_) are also influenced by the sequence of amino acids, so that, depending on the identities of the amino acids immediately on either side of it, a particular backbone amide can have a *k*_ch_ that varies by up to two orders of magnitude. The dependency of the HX rate on sequence has been thoroughly characterized by Englander and others [8,9]

In folded proteins, the observed exchange rate is modulated by structural features. Backbone amide hydrogens that are hydrogen bonded, for instance, like in α-helices or β-sheets, do not undergo exchange with solvent. Similarly, amide hydrogens that are located in a solvent-inaccessible region of the protein do not undergo an exchange with the solvent. However, while a small fraction of backbone amides are so strongly hydrogen bonded or sequestered that they essentially never undergo an exchange with the solvent, the vast majority will become transiently available through thermally driven structural fluctuations (i.e., conformational dynamics) in which hydrogen bonds are transiently broken and/or ‘unexposed’ regions become transiently exposed. This transient availability is the basis for HDX as a measure of conformational dynamics in proteins. The basic model for exchange occurring at an individual backbone amide in a folded protein can be expressed as follows [11,12]:(2)$${\mathrm{NH}}_{\mathrm{cl}}\overset{k_{\mathrm{op}}}{\underset{k_{\mathrm{cl}}}{⇌}}{\mathrm{NH}}_{\mathrm{op}\;}\overset{k_{\mathrm{ch}\;}}{\to }{\mathrm{ND}}_{\mathrm{op}\;}\overset{k_{\mathrm{cl}}}{\underset{k_{\mathrm{op}}}{⇌}}{\mathrm{ND}}_{\mathrm{cl}\;}$$
where cl represents the closed (unavailable for exchange) conformation, op corresponds to the open (available for exchange) conformation, and *k*_op_ and *k*_cl_ represents the rate constants for the opening and closing reactions.

According to this model, the observed exchange rate constant *k*_HX_ is affected by the rate of opening and closing transition and the chemical exchange rate (*k*_ch_). Therefore, the observed rate constant for HX exchange under native conditions can be expressed as [13]:(3)$$k_{\mathrm{HX}}=\frac{k_{\mathrm{op}\;}\times k_{\mathrm{ch}}}{k_{\mathrm{cl}}+k_{\mathrm{ch}}}$$

Depending on the relative values of *k*_cl_ and *k*_ch_, amide exchange MS profile can be classified into two different regimes: EX1 and EX2. In the EX2 regime, the rate of closing is much faster than the chemical exchange rate (*k*_cl_ >> *k*_ch_). Therefore, the amide must undergo many brief unfolding events before the hydrogen gets exchanged with deuterium. The EX2 regime is dominant for protein HDX under ‘native-like’ conditions, and by MS it is characterized by a gradual shift in the peptide isotopic distribution (Figure 2). It is important to recognize that, in this regime (i.e., where *k*_ch_ is very small relative to *k*_cl_), expression (3) simplifies to *k*_HX_ = *k*_ch_(*k*_op_/*k*_cl_). Since *k*_ch_ can be calculated based on pH and primary sequence, the only unknown in this expression is the equilibrium constant for the closed → open transition (*k*_op_/*k*_cl_) = K_op_. This equilibrium constant is sometimes inverted and called the ‘protection factor’, since 1/K_op_ is effectively a measure of the degree to which structure ‘protects’ the amide from undergoing exchange. It is also is also sometimes used as a quantitative measurement of local structural stability (i.e., ΔG unfolding) at specific sites on the protein [14,15,16], although such quantitative interpretations should be taken with a high degree of caution. EX1 kinetics are observed when the rate of an opening/closing transition is lower than the chemical exchange rate (*k*_cl_ << *k*_ch_). In this case, each opening event of amide results in a complete exchange of hydrogen with deuterium and shows a characteristic bimodal isotopic distribution [17] (Figure 2).

## 2. HDX-MS Methodology

There are two conventional workflows for HDX-MS experiments: the one most commonly used is the ‘bottom-up’ approach where the deuterium-labeled protein is digested by pepsin and peptides are analyzed by mass spectrometry, and the second is the ‘top-down’ approach where the protein is ionized intact and then fragmented inside the mass spectrometer. The bottom-up approach involves the following steps (1) HDX labeling, (2) acid quench, (3b) pepsin digestion, (4) LC separation and electrospray MS and (5) data analysis (Figure 3). The top-down method involves (1) HDX labeling, (2) acid quenching, (3b) electrospray MS and in-MS fragmentation via non-ergodic process and (4) data analysis (Figure 3).

### 2.1. Labeling of Protein

The HDX labeling step can be carried out in several different ways depending on the nature of the experiment. The most common method used is simply to dilute the protein sample (in buffered H_2_O) into an analogous D_2_O buffer. The main disadvantage of this method is that the protein sample gets diluted, which can affect the results when studying the protein–protein interactions of weak binders. Fortunately, though it is not widely recognized by those carrying out HDX, the conventional 10X dilution factor is entirely unnecessary in HDX experiments, provided that the lower dilution level is accounted for in the analysis [18]. In fact, it is, in many cases, possible to acquire reliable HDX measurements from mixtures that contain only 10% D_2_O [18]. The next method is the lyophilization of the protein sample and then resuspending into D_2_O buffer. This approach has been successfully used to study the self-association of antibodies at a high concentration [19], but the methodology has a limitation in that all proteins are not amicable to lyophilization without substantial structural perturbation. Another recent approach is microdialysis of protein sample in D_2_O buffer [20]. The advantage of this method is the same as earlier, but this method requires a large volume of D_2_O buffer, which can be expensive.

In conventional HDX experiments, the labeling step is carried out for increasing lengths of time, ranging from 10 s to several hours. The number of timepoints collected can vary, but the minimum requirement is typically four, spanning the seconds-to-hours timescale.

### 2.2. Quenching and Digestion

After labeling of the protein, the reaction is quenched by reducing the pH to 2.5 and temperature to 0 °C. Since the H/D exchange reaction is more efficiently base catalyzed than acid catalyzed, lowering the pH to 2.5 effectively ‘freezes’ the label, reflecting the structural and dynamic characteristics of the protein under ‘native-like’ conditions (prior to the pH drop). This low pH greatly reduces the back exchange of deuterium with hydrogen in all subsequent steps. Back exchange corrections must be performed using model peptides. These model peptides are exposed to D_2_O buffer until saturated and analyzed for back exchange for each unique workflow. The percentage of back exchange for the model peptides is then calculated, and experimental peptides corrected to reflect their true uptake values [21,22]. To aid the proteolytic digestion, the quench buffer often contains a denaturing agent such as GdmCl and a reducing agent like TCEP to reduce the disulfide bonds [23]. To digest the protein, most widely used protease is pepsin, but other proteases like Protease XIII are also used in some specific experiments [24,25]. In a conventional setup, the protein sample is passed through an immobilized protease column (~10 to 15 °C), and the efficiency of protease digestion can be modulated by varying the flow rate of the mobile phase.

### 2.3. Separation of Peptides and Mass Spectrometry

Peptides generated by the protease column are separated by analytical HIC columns (C18 or C8) maintained at low temperature on HPLC/UPLC systems. Separation is carried out over a short time (usually 5–15 min) to minimize the back exchange (which still occurs slowly despite the low pH). With the advancement of technology, robotic systems are available for performing labeling, quenching, and sample injection [26]. The purified peptides are then injected into the mass spectrometer using an electrospray ionization source, in a manner that is analogous to classical LC–MS. This process is repeated for each labeling timepoint, resulting in a set of LC–MS chromatograms that correspond to the set of peptides at different labeling times. As labeling times increase, peptides with exchangeable sites will exhibit shifts in their isotopic distributions, resulting from the progressive uptake of the deuterium label.

### 2.4. Data Analysis

A detailed discussion of the principles underlying HDX-MS data analysis and interpretation is beyond the scope of this review. For a thorough exploration of this topic, the reader is referred to refs [27,28,29]. Over the past few years, a number of automated HDX-MS data analysis software suites have become available, including MS Studio, HDX Workbench [30], DECA [31], HD desktop [32], Hydra [33] and ExMS [34], which have had a dramatic impact on the accessibility of the method. However, there remain significant challenges with each of these software solutions, and close manual supervision of the automated process is recommended.

### 2.5. Millisecond Bottom-Up HDX-MS

In addition to bottom-up HDX on the conventional timescale, where the shortest labeling time is typically 10 s, a small number of groups have developed methods for implementing the bottom-up HDX workflow with millisecond-to-second labeling times [35,36,37]. One implementation incorporates labeling, quenching and proteolysis steps into a single microfluidic device with labeling times from 80 ms to 10 s achieved using an adjustable capillary mixer [38,39,40,41]. When millisecond labeling times are used, deuterium uptake is being monitored on a similar timescale to the relaxation times of most chemical exchange processes (i.e., 1/kch) at neutral pH. This makes millisecond HDX uniquely sensitive to sites where the backbone amide exchange is only slightly attenuated by K_op_, which occurs in weakly structured regions of proteins, intrinsically disordered proteins, and weak (or rapid turnover) protein/ligand interactions. Short labeling times are also more sensitive to subtle changes in conformational dynamics in ‘comparative’ HDX analyses, e.g., when the dynamics of a ligand-free and ligand-bound are being compared [42].

### 2.6. Top-Down HDX-MS

Top-down HDX is far less common than bottom-up HDX-MS, in large part due to the issue of ‘scrambling’ that occurs with the most commonly used fragmentation techniques. Scrambling involves the migration of protons and/or deuterium along the peptide backbone as the protein ion is heated (through collisions with neutral gas) prior to fragmentation, which can result in the complete loss of the original labeling pattern. Some less commonly used fragmentation techniques, including Electron Capture Dissociation (ECD) and Electron Transfer Dissociation (ETD) are ‘non-ergodic’ in the sense that they occur in a single, rapid step [43,44]. These can be carried out without scrambling; however, they have their own limitations including fragmentation inefficiency and an inability to separate the peptides that are generated, resulting in exceedingly complex mass spectra, particularly from large proteins [45]. A significant advantage of top-down HDX-MS is that it is the only MS-based approach that can generate true site-specific HDX measurements [46].

## 3. Applications of HDX-MS

Various applications of HDX-MS are outlined in Figure 4, and are described in detail in the following sections.

### 3.1. Protein Conformation and Comparability

The classic use of HDX-MS is to compare different conformers of a protein. In many diseases, a single base variation in DNA sequence causes an amino acid substitution, affecting the protein structure and stability. Many recent examples of HDX-MS structural comparisons are derived from the amyloidosis literature, where point mutations can induce shifts to pathogenic, non-native structures (or ensembles). Prion diseases are a group of neurodegenerative disorders caused by the misfolding and aggregation of the Prion (PrP) protein. A single-residue mutation A116V in PrP enhances the formation of a neurotoxic form called the transmembrane form, which induces neural death. Sabareesan and Udgaonkar compared the deuterium uptake of the wild type with the mutant protein and observed that the peptide (109–132) responsible for membrane insertion is more sequestered from the solvent in the mutant protein, causing the oligomerization of prion protein on the membrane surface and inducing pore formation activity (Figure 5) [47].

In a set of linked studies, human immunodeficiency virus 1 (HIV-1) was studied using HDX-MS due to its incompatibility with other structural techniques, as it is aggregation prone and can only be isolated in small quantities. HIV-1 survives inside the host cell by suppressing the antiviral activity of the APOBEC3 family of enzymes. HIV viral infectivity factor (Vif) is the protein responsible for the inhibition of the host cell. Engen and coworkers have used HDX-MS to probe the protein conformation and provide new insights into the solution structure of Vif protein, as well as its complexation with its Elongin BC protein binding partner [48]. Vif alone displays low deuteration levels in the N-terminal region of the protein, suggesting that it is protected, either due to secondary structural elements or sequestration, from the solvent. In contrast, high deuteration levels in the C-terminal region suggests that this region is likely unstructured in the absence of cellular interacting proteins (Figure 6a). When complexed with its Elongin BC binding partner, Vif induces significant stabilization, as localized peptides show decreased levels of deuterium uptake [49] (Figure 6b). In addition, the study was able to confirm that Vif binds to Elongin BC without the presence of other cellular factors, making it a promising target for anti-HIV-1 drug design. The authors note that HDX-MS can be a useful assay tool for the screening of Vif:Elongin BC inhibitors.

### 3.2. Protein–Small Molecule Binding

Another important application of HDX-MS is the study of protein conformational or dynamic change induced by binding to a small ligand. The conformational response of a protein to a small ligand or drug molecule binding is not always straightforward. The binding of a ligand can cause both orthosteric and allosteric changes in a protein molecule, making it difficult to discern the binding site from the allosteric site. HDX-MS is an invaluable orthogonal tool that can detect these changes, which often go undetected from X-ray crystallography studies alone. Apolipoprotein D (ApoD) is a secreted glycoprotein and is found to be associated with various neurological disorders such as Alzheimer’s and Parkinson’s diseases [50]. Progesterone is a common hydrophobic ligand for ApoD and the crystal structure of free and progesterone-bound states of the protein do not show any conformational changes. However, HDX-MS studies on the free and progesterone-bound state of the ApoD showed significant conformational differences between the two states. In the progesterone-bound state of ApoD, both the binding site pocket and distal regions at the N and C-terminal end are more stabilized [51]. The only region showing increased exchange spans amino acid residues 49–61, which contains Asn−58 of the binding pocket (Figure 7). The localized solution phase information obtained from HDX-MS can help make ApoD an important therapeutic target for several devastating neurological diseases. An excellent review on the use of HDX-MS for studying protein–small molecule interactions in the context of drug development is referenced for the reader’s interest [52].

### 3.3. Epitope Mapping

Epitope mapping, the process of identifying the antibody binding region on the antigen surface, is one of several rapidly growing industrial (pharmaceutical) applications of HDX-MS [3,53,54,55,56,57]. Through epitope mapping, HDX-MS provides valuable insight into the mechanism of different antibodies binding to the same antigen in the solution phase. In a recent study by Zhu et al., HDX-MS studies performed on two different monoclonal antibodies (mAbs), 2–25 and 2–18, binding to the diphtheria toxin (DTx) antigen revealed two different binding sites on the antigen [58]. Briefly, mAb 2–25 selectively bound to the B-subunit of DTx containing the translocation and receptor domain, in turn blocking the heparin-binding EGF-like growth factor-binding site. On the other hand, mAb 2–18 binds to the catalytic domain of the A-subunit that shuttles NAD during catalysis (Figure 8). These different mechanisms help shed light on the efficacy of the DTx vaccine, as one mAb inhibits cell recognition and the other catalysis. Rapid and robust characterization of these antibody–antigen interactions can further our knowledge of the fundamental biological function of a given antigen, as well as help in the reagent selectivity process of vaccine development.

### 3.4. Multi-Protein Complexes

A particular strength of HDX-MS is the structural characterization of large or multi-protein complexes. Many cellular processes require assembly or interactions between multiple proteins, and the study of large complexes by standard methods, such as X-ray crystallography, and NMR is often prohibitively challenging. With advancements in LC systems and mass spectrometer resolution, it is now possible to characterize large proteins and protein complexes using bottom-up HDX-MS. A good example of this is a recent study by Vahidi et al., where the authors were able to obtain conformational dynamics information associated with the catalytically active FOF1 ATP synthase complex [59]. This 525 kDa complex consists of various membrane-embedded subunits responsible for both synthesis and hydrolysis of ATP. Using *E. coli* vesicle membrane mimics, three catalytic conditions were examined: 1) an ADP-bound inhibited state, IADP, 2) a catalytically active state where the inner-membrane proton-motive force (PMF) is high, WPMF, and 3) a working state where carbonyl cyanide−4-(trifluoromethoxy) phenylhydrazone (FCCP) prevents proton build-up in the membrane, WFCCP (Figure 9a). Comparing the deuterium uptake profiles from all three states can allow for mechanistic insights into how these biological motors carry out essential functions. For example, the authors noted that the C-terminal helix peptide of the γ subunit (260LQLVYNkARQASITQEL276) in the WPMF state displays destabilization during catalytic turnover. One explanation for this could be that the γ sub-unit exerts mechanical stress on the α3β3 sub-unit, and when under a PMF load, the C-terminal portion of the γ sub-unit over-twists, causing hydrogen bonds to destabilize (Figure 9b-d). Another study performed HDX-MS using a modified nanospray configuration to characterize DNA-PKcs, a 469 kDa protein, with 90% sequence coverage [60]. This configuration can be particularly useful for the study of proteins that have a higher complexity, and that cannot be produced in large quantities.

### 3.5. Intrinsically Disordered Proteins

Intrinsically disordered proteins (IDPs) lack unique 3D structures and adopt a plethora of highly dynamic conformations in solution. Although IDPs are disordered in the native structure, they can form local transient secondary structures known as residual structures [61]. IDPs or intrinsically disordered regions (IDRs) in proteins play an essential role in numerous biological processes such as cell signaling, gene regulation, and posttranslational modification. The highly dynamic nature of IDPs complicates the crystallization of protein, making NMR and HDX-MS a valuable tool to study the IDPs. However, NMR requires a high concentration of protein samples, which can affect the stability of many IDPs. Therefore, HDX-MS can be a useful tool to study the conformational dynamics of IDPs. The conformational transitions of IDPs can only be detected with millisecond labeling where a short labeling time in hundreds of milliseconds can detect transiently populated states of IDP. Recently, Zhu and coworkers reported millisecond HDX-MS experiments on the tau protein, an amyloidogenic intrinsically disordered protein. The study provided a detailed picture of the residual structure in tau compared to its hyperphosphorylated form, which is shown to be more amyloidogenic in nature [62]. Tau contains a microtubule-binding region, as well as two hexapeptide motifs, 275VQIINK280 (H1) and 306VQIVYK311 (H2), which have been linked to amyloidogenic propensity. The study found that hyperphosphorylation results in the release of the global fold, the development of new intramolecular interactions in the microtubule-binding region and increased exposure of H2 (Figure 10). Taken together, this study successfully highlights the specific structural changes that occur as tau transitions to its pathogenic state, making time-resolved HDX-MS useful in the study of challenging IDPs.

### 3.6. Membrane Proteins

Membrane proteins are among the most challenging class of proteins to study in structural biology. They nonetheless constitute an important group of proteins required for molecular transport, signal transduction, and enzyme reactions. Moreover, membrane proteins are the most targeted molecule in drug discovery. Therefore, understanding the conformational dynamics of membrane proteins is particularly crucial for drug design. One of the major challenges of studying membrane proteins is the stability of the protein in solution. Membrane proteins are rich in hydrophobic residues that tend to aggregate in solution, so they always need to be stored in either detergents or liposomes or micelles. Traditional structural biology tools such as X-ray crystallography or NMR are largely not compatible with membrane proteins because these techniques require a large amount and high purity, a soluble protein which is difficult to obtain for membrane proteins. HDX-MS has been extensively used to characterize G-protein coupled receptors (GPCRs) and the bound state component. GPCRs are involved in cell signal transduction and is an important target molecule in the drug development industry. HDX-MS provides valuable insight into the conformation of the GPCRs when bound to different ligands [63]. For example, one of the pioneering studies which applied HDX-MS to a member of the GPCR protein family took place in 2010, where Zhang et al. studied detergent solubilized β2-Adrenergic GPCR (β2AR) alone and bound to the inverse agonist carazolol [64]. To obtain high sequence coverage, many parameters were optimized, including detergent amounts, quench solution composition, and LC parameters. In a later study performed by Duc et al., the conformational dynamics of three GPCRs were compared, including β2AR, reconstituted in dodecyl maltoside (DDM), and lipid bicelles [65]. The authors found that solubilization in bicelles improved the sequence coverage of β2AR by almost 20% (Figure 11).

In recent years, the use of nanodiscs has gained popularity due to their increased stability compared to liposomes or micelles. Nanodiscs consist of a lipid bilayer held together by a protein belt and can be highly customizable to the size of the protein under study [66]. Although HDX-MS can be employed to study membrane proteins, the presence of detergents, lipids, and nanodiscs affects the performance of both LC and MS systems. The presence of a pre-guard column removes most of the detergents, but still, carryover can be seen in the mass spectrometry. Therefore, the removal of lipids and nanodiscs must be performed using zirconium oxide beads prior to analysis. Another challenge in the use of HDX-MS is the presence of a lot of hydrophobic residues in the membrane protein, which can cause carryover into the next sample. Such issues can be addressed by employing different analytical tools such as C8 instead of C18. A detailed discussion of the sample preparation and mass spectrometry parameters specifically used for the study of membrane proteins can be found in several recent reviews [67,68,69].

### 3.7. Glycosylated Proteins

Glycoproteins play a highly diverse function in cells, ranging from cell–cell recognition and immune function to pathogen identification. The study of glycosylated proteins with HDX-MS involves several challenges such as the limited digestion of protein due to steric hindrance from glycans or difficulty in the identification of peptides due to the heterogeneity in the glycan composition. Due to the hydrophilic nature of glycans, they can also interfere with the binding of peptides to the C18 column. Recently, Jensen et al. reported a PNGase A enzyme that can catalyze the deglycosylation of proteins under an acidic pH and a low temperature [70]. The incorporation of this enzyme in the quench buffer can remove the glycan chains and reduce the complexity of the protein sample. The monoclonal antibody Trastuzumab was used as a test subject for this revised HDX-MS workflow. The disulfide bonds in Trastuzumab were reduced using 0.5 M TCEP added to the quench buffer. They were able to detect the deglycosylation of N-linked glycopeptides and compare it to the native state, highlighting the structural changes induced by glycosylation (Figure 12). HDX-MS can also be used to detect how different glycosylations can impact the overall structure of a protein. One recent study was able to distinguish the subtle changes that occur in the idursulfase enzyme when glycosylation sites were modulated [71]. This can have significant applications in the pharmaceutical industry, as monoclonal antibodies produced by different manufacturing processes often have differing glycosylation patterns.

## 4. Conclusions

HDX-MS is a rapidly growing in structural biology, and automation of commercial HDX systems, together with streamlining of data analysis, provides an opportunity to do high-throughput screening of drug molecules as well as protein therapeutics. The versatile nature of this technique can be used to enable the study of folding intermediates or rarely populated species in proteins, together with large macromolecular assemblies. This is not to say that HDX-MS does not have significant challenges that will need to be overcome before it can be fully adopted in academia and industry. A large number of factors such as temperature, pH (pD), and the ionic strength of the buffers can affect the percentage of deuterium uptake for particular peptides in unique ways. While these factors can be partially controlled using calculated intrinsic rates of exchange, data reproducibility remains one of the most pressing challenges in the field [21]. Furthermore, pepsin tends to generate different sets of peptides for the same protein, even under highly similar experimental conditions, which makes data comparison among multiple labs difficult. To address the issue of data reproducibility, a large number of research groups use an automated sample dispensing system or internal standards to correct any offsets in the data due to any of the factors mentioned earlier [22,72,73]. Another challenge for the HDX-MS community is to develop a concise and effective mode of data presentation and interpretation. Each HDX-MS experiment generates an enormous amount of data, which should be processed and curated in such a way that the data can be easily comprehensible, both by reviewers and the scientific community.

Challenges aside, HDX-MS is an approach that provides valuable insights into some of the most challenging biological targets, such as membrane proteins, disordered proteins, and glycosylated proteins. With the commercialization and advancement of the technique, systems will become more user friendly so that the broader scientific community, including industrial researchers, can access and apply the technique to their scientific studies.

## Figures and Tables

**Figure 1 biomedicines-08-00224-f001:**
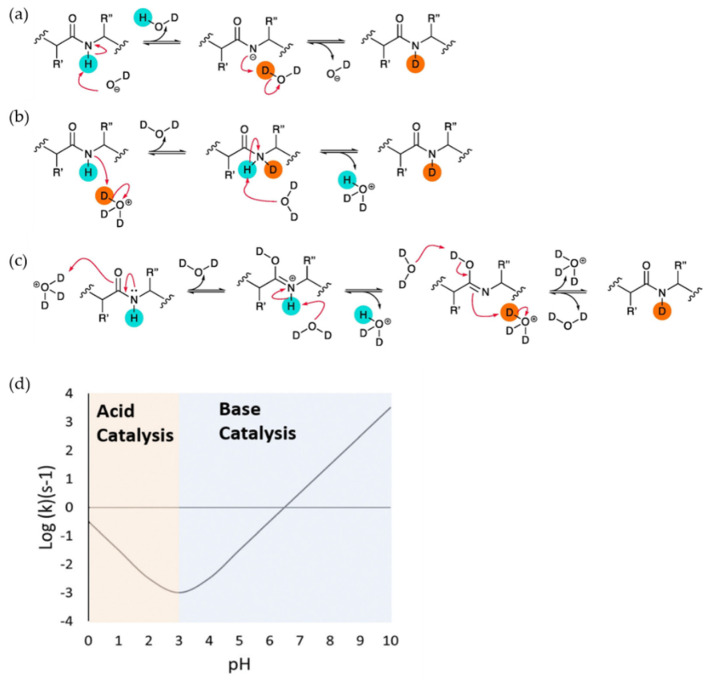
Schematic of acid and base catalysis of peptide group protons in solution during H-D exchange and the dependence of intrinsic rate of exchange (kch) on the pH of solution. (**a**) Base catalysis (**b**) Acid catalysis through the protonation of peptide group N atom. (**c**) Acid catalysis through protonation of peptide group O atom. Reproduced with permission from Journal Methods@ 2018. (**d**) The dependence of intrinsic rate of exchange (kch) of poly-DL-alanine on the pH showing that the minimum exchange is at pH 2.5–3. Acid catalysis is dominant below the pH and base catalysis is dominant above the pH. Reproduced with permission from the journal *Analyst* (© 2017)**.**

**Figure 2 biomedicines-08-00224-f002:**
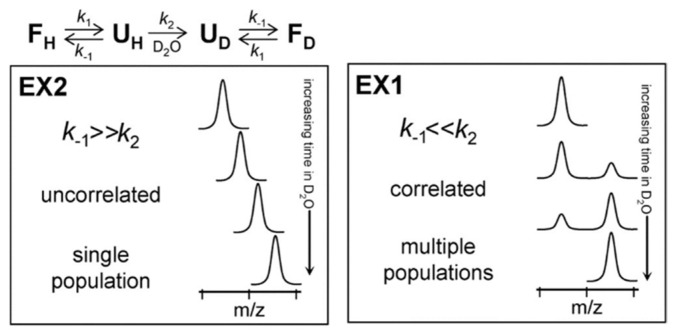
Protein unfolding kinetics and Hydrogen–Deuterium Exchange Mass Spectrometry (HDX-MS) at the top. The two possible mechanisms for hydrogen exchange in proteins, EX1 and EX2. Reproduced with permission from the *Journal of the American Society for Mass Spectrometry* (© 2006).

**Figure 3 biomedicines-08-00224-f003:**
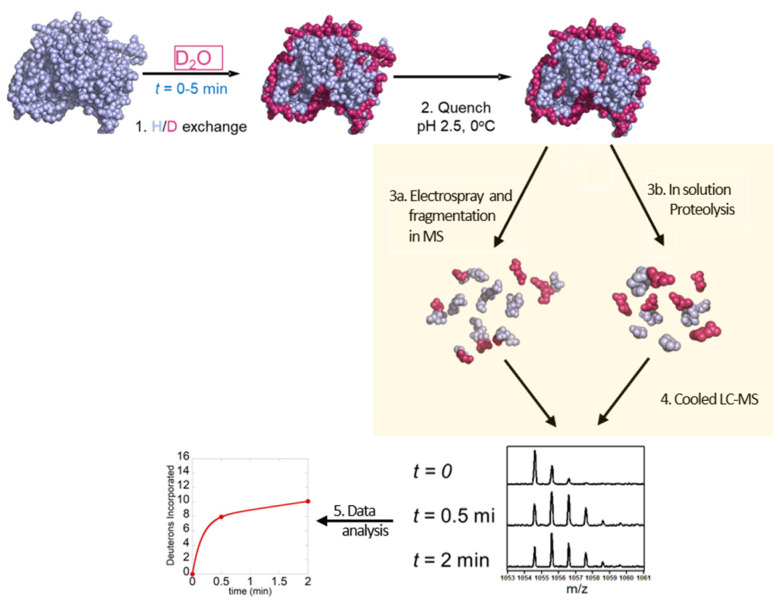
Schematic overview of top-down and bottom-up HDX-MS workflow: 1—exchange of hydrogen to deuterium for different time points (0–5 min), 2—quenching at low pH, 3a—proteolysis, 3b—electrospray and fragmentation in MS 4—separation of peptides by LC–MS and 5—data analysis.

**Figure 4 biomedicines-08-00224-f004:**
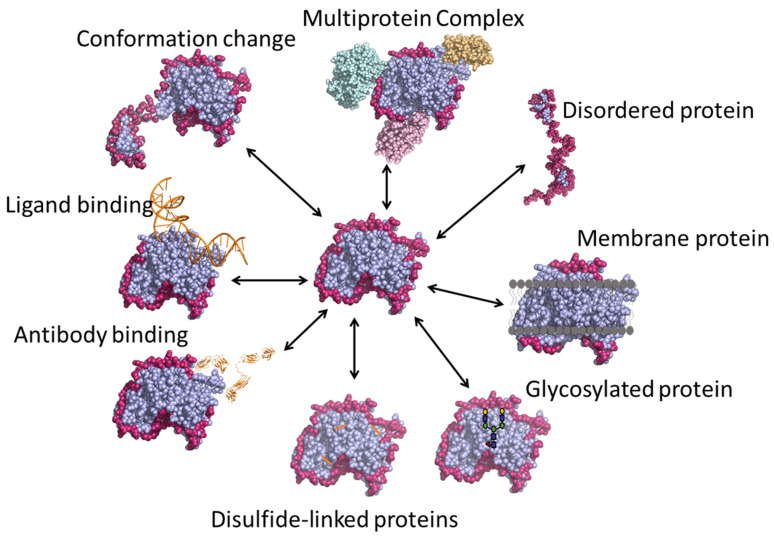
Various applications of HDX-MS technique.

**Figure 5 biomedicines-08-00224-f005:**
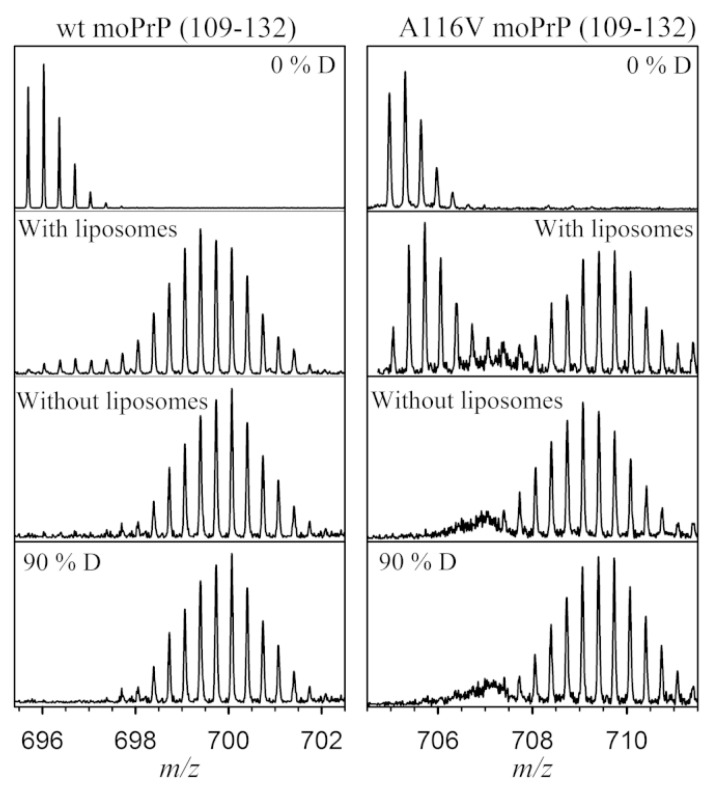
The peptide spanning 109–132 in WT and A116V moPrP displays a significant decrease in deuterium uptake in the presence of liposomes. At 30 s of deuterium labeling, there is a bimodal distribution in the presence of liposomes, compared with the unimodal distribution seen in the absence of liposomes. The controls of protonated (0% D) and deuterated (90% D) peptide fragments are also shown. Reproduced with permission from the Biophysical Society (© 2016).

**Figure 6 biomedicines-08-00224-f006:**
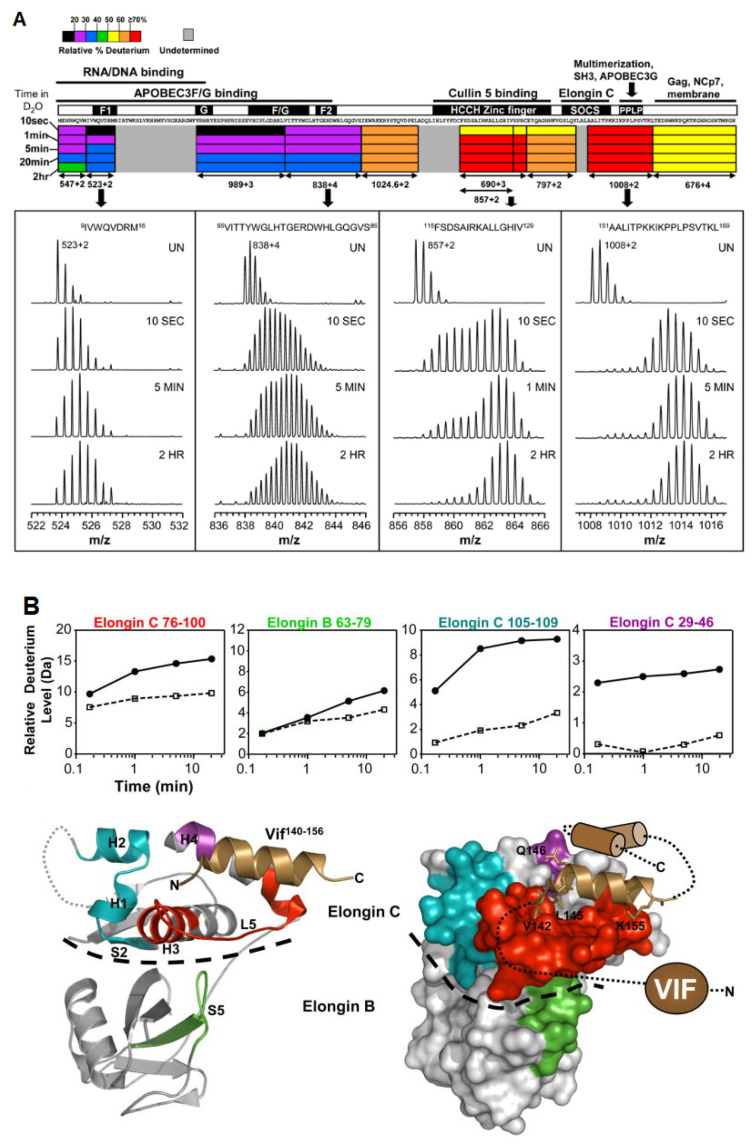
(**A**) The deuteration map for recombinant Vif. The bars are colored according to the relative percent deuterium incorporation as indicated in the legend. Additionally indicated are the different regions of Vif and their corresponding biological functions. Representative spectra of four peptides are shown, with deuteration times included in each panel; UN represents an undeuterated protein. Reproduced with permission from Elsevier Ltd. ©2011. (**B**) Location of conformational changes in Elongin B and C induced by HIV-1 Vif interaction. Deuterium uptake curves for Elongin B and C peptides that displayed changes in deuterium uptake upon interaction with Vif are shown in the top panel. The solid lines represent the Elongin BC complex alone, and the dotted lines represent the Elongin BC complex bound to Vif. The changes are mapped onto the structure of the Vif_140–156_. Elongin BC complex. The colors in the top panel correspond to those mapped onto the structures. Areas that are grey indicate regions with no change in deuterium uptake or those which were not covered. The HIV-1 Vif peptide 140–156 is represented as the copper helix, and the remaining structurally uncharacterized regions of Vif are depicted in cartoon form. A dotted line indicates the boundary between Elongin C and Elongin B. Reproduced with permission from Elsevier Ltd. (© 2010).

**Figure 7 biomedicines-08-00224-f007:**
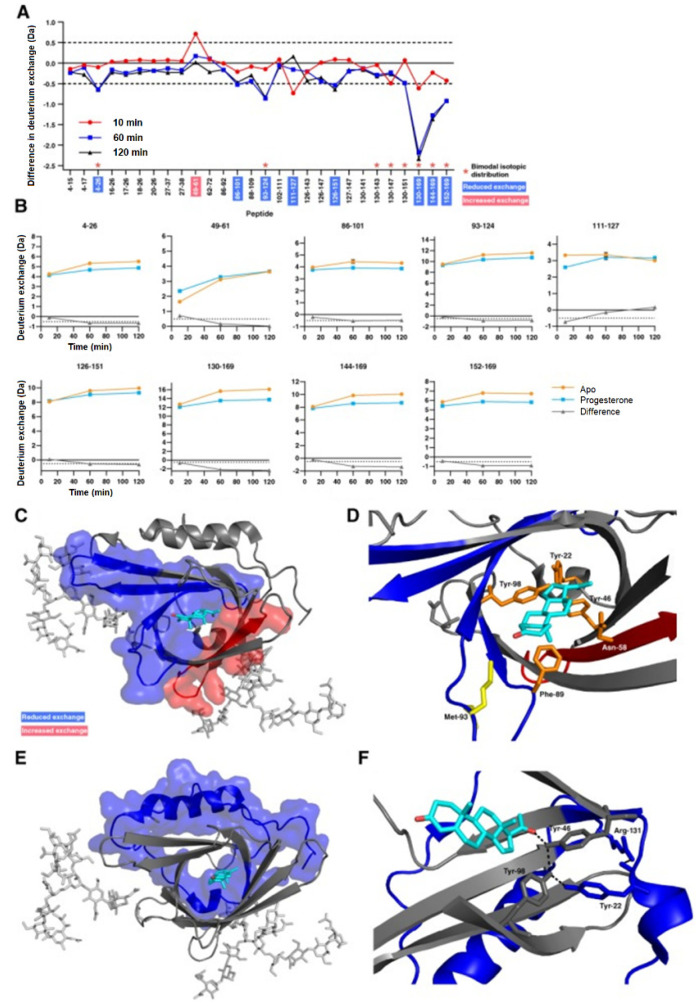
(**A**) Difference plot generated by subtracting the absolute deuterium exchange of ligand-free ApoD from progesterone-bound ApoD. Differences > ±0.5 Da are considered significant and were found in nine peptides. Asterisks denote peptides showing bimodal exchange profiles in both the apo-form and in the presence of progesterone. (**B**) Absolute deuterium exchange over time of nine peptides showing significant changes upon progesterone binding. (**C**) Significant orthosteric changes in ApoD upon progesterone binding. (**D**) Zoom of the ligand binding pocket. Asn−58 is located in peptide 49–61 which shows increased deuterium exchange upon progesterone binding. Met−93 and Phe−98 are located in peptides which show decreased deuterium exchange upon progesterone binding. (**E**) Significant allosteric changes in ApoD upon progesterone binding. (**F**) Hydrogen bond network between progesterone, Tyr−46, Tyr−22, Tyr−98, and Arg−131. Reproduced with permission from The Protein Society (© 2018).

**Figure 8 biomedicines-08-00224-f008:**
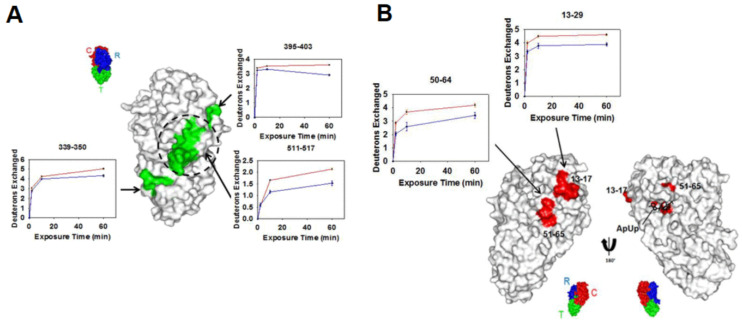
(**A**) Regions showing decreases in deuterium uptake upon diphtheria toxin (DTx)- monoclonal antibody (mAb) 2–25 complexation are shown in green and mapped onto the structure of DTx. The respective kinetic plots are also shown, with red and blue plots representing the uptake in the absence and presence of mAb 2–25, respectively. A reference structure with labeled domains is shown in the inset, with the dashed circle representing the HBEGF–binding pocket. (**B**) Regions showing decreases in deuterium uptake upon DTx-mAb 2–18 complexation are shown in red and mapped onto the structure of DTx. The respective kinetic plots are also shown, with red and blue plots representing the uptake in the absence and presence of mAb 2–18, respectively. Reproduced with permission from the American Chemical Society (© 2018).

**Figure 9 biomedicines-08-00224-f009:**
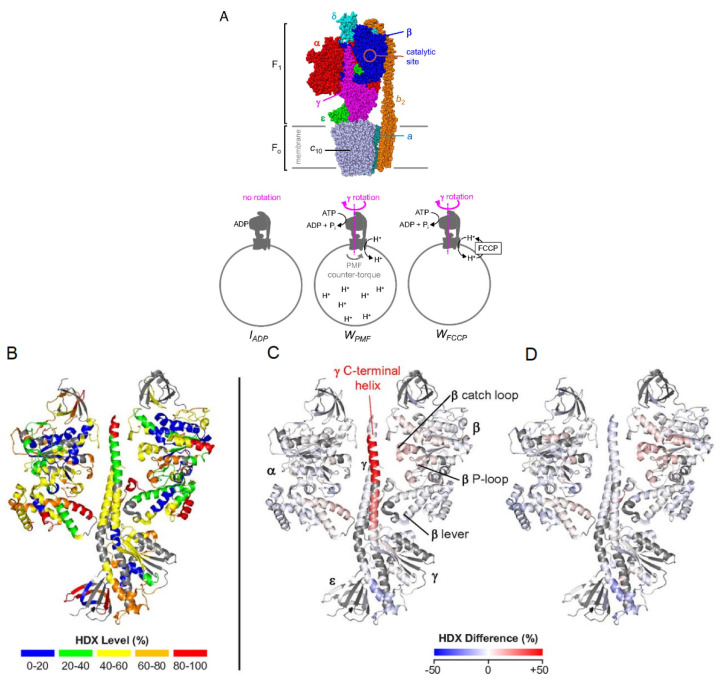
(**A**) Subunit architecture of F_O_F_1_ ATP synthase from E. *coli*. The bottom panel depicts membrane vesicle-bound F_O_F_1_ under different experimental conditions. (**B**) HDX levels of *I_ADP_* for an HDX period of 45 min. (**C**) Deuteration difference map of *W_PMF_* vs. *I_ADP_*. (**D**) Deuteration difference map of *W_FCCP_* vs. *I_ADP_*. Regions that were not covered are shown in dark gray. Colors represent deuteration percentages or HDX difference (%) as indicated in the legends. Reproduced with permission from the National Academy of Sciences (© 2016).

**Figure 10 biomedicines-08-00224-f010:**
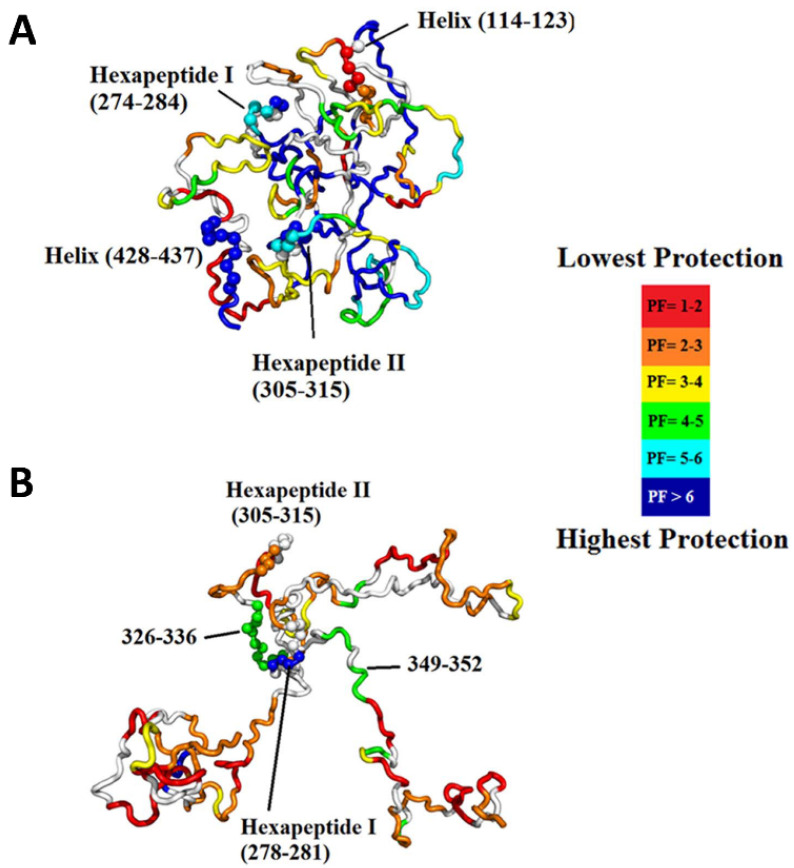
(**A**) Most representative structure for the native ensemble, which exhibits a global fold with sequestration of the hexapeptides. The structure is colored based on protection factor, as indicated in the legend. (**B**) Most representative structure for the hyperphosphorylated ensemble showing release of the N- and C-termini containing the microtubule-binding region, full exposure of H2 and a few regions of residual structure, including H1. Reproduced with permission from PLoS ONE (© 2015).

**Figure 11 biomedicines-08-00224-f011:**
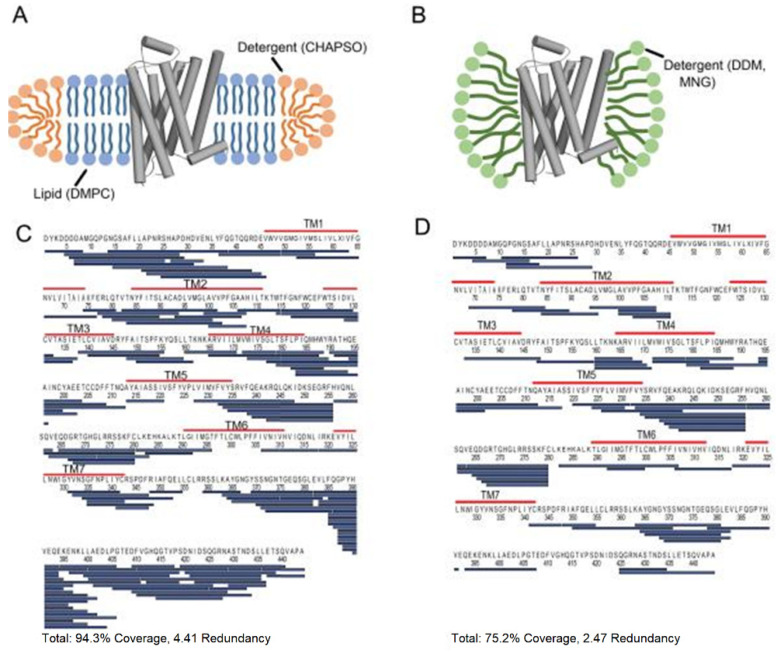
Cartoon illustration of β2-Adrenergic G-protein coupled receptor (GPCR) (β2AR) in (**A**) a lipid bicelle (3:1 DMPC:CHAPSO) and (**B**) dodecyl maltoside (DDM). (**C**) The sequence coverage of β2AR prepared in bicelles and (**D**) DDM. Blue bars represent peptic peptides identified from four independent experiments. Reproduced with permission from the American Chemical Society (© 2015).

**Figure 12 biomedicines-08-00224-f012:**
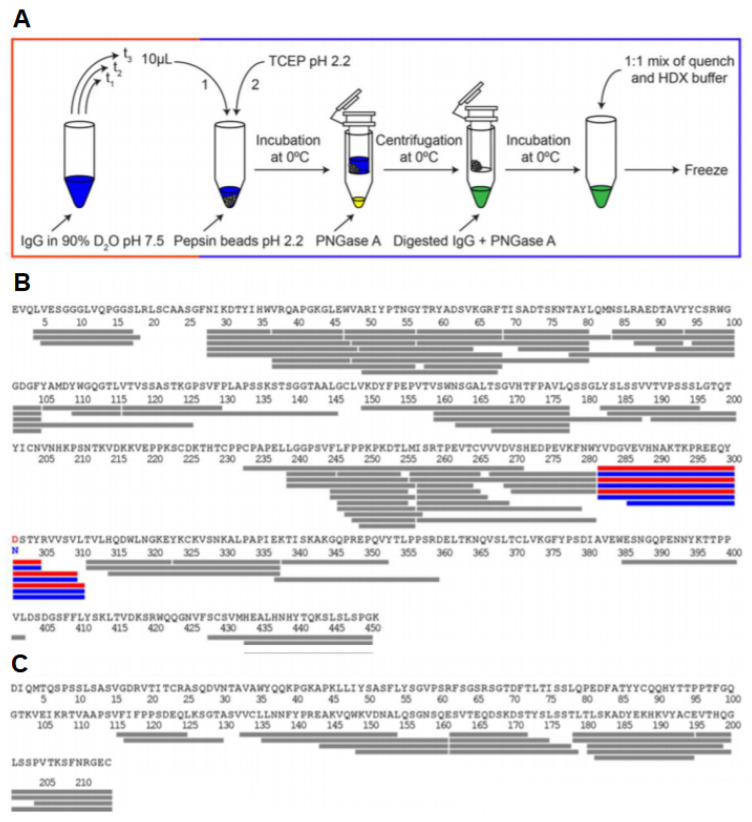
(**A**) HDX-MS workflow with in-solution deglycosylation at low pH using PNGase. HDX-MS sequence coverage of Trastuzumab (**B**) heavy chain and (**C**) light chain using the optimized workflow. The glycosylated residue N301 and the deamidated deglycosylated residue D301 are shown in blue and red, respectively, along with the detected N-linked glycopeptides. Reproduced with permission from the American Chemical Society (© 2016).

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
