# Peer review of "HDX-MS: An Analytical Tool to Capture Protein Motion in Action"

_biomedicines, 2020, doi:10.3390/biomedicines8070224_

Round 1

Reviewer 1 Report

 Narang et al reviewed the principles, methodology and examples of application of Hydrogen-Deuterium Exchange Mass Spectrometry.

The review is well written and the material described logically and clear. The figures support all authors statements.

Therefore, I recommend that the review will be accepted.

The minor comment is that it will be nice if Figure 3 will include the scheme of top-down methodology for more straight comparison.

Author Response

We would like to thank the reviewers for their careful and constructive review of the manuscript. Detailed responses to reviewer comments and accompanying changes are provided below (changes are noted in italic). We hope that you will find the accompanying revised version of the manuscript suitable for publication in Biomedicines.

Reviewer 1:

The minor comment is that it will be nice if Figure 3 will include the scheme of top-down methodology for more straight comparison.

We have included the top down mechanism in Figure 3.  

Reviewer 2 Report

This is an excellent review on the application of H-D exchange mass spectroscopy (HDXMS) to detect protein dynamics.  It is very well organized, beautifully illustrated and contains up to date references  covers the various aspects of HDXMS as it pertains to protein dynamics. These include, an in depth review of the methodology, applications- protein conformation, protein-protein interactions (and multi protein interactions), protein-small molecule interactions, epitope mapping, intrinsically disordered proteins, membrane proteins and disordered proteins.

One suggestion to improve the review would be to add a figure in the introduction section to illustrate acid and base catalyzed HDX.

Author Response

We would like to thank the reviewers for their careful and constructive review of the manuscript. Detailed responses to reviewer comments and accompanying changes are provided below (changes are noted in italic). We hope that you will find the accompanying revised version of the manuscript suitable for publication in Biomedicines.

Reviewer 2:

One suggestion to improve the review would be to add a figure in the introduction section to illustrate acid and base catalyzed HDX.

We have included a new figure in the introduction section explaining the acid and base catalyzed HDX reaction

Reviewer 3 Report

This review article provides a survey on the use of HDX-MS for the characterization of protein dynamics. It gives an overview of the applications of various protein modalities, including complexes and protein ligand interactions, with each subsection in section 3 providing one main example for each modality. While there are many reviews already published on this topic, this survey approach offers a distinct perspective.

I have several comments and suggestions that need to be addressed:

  • The last sentence of the abstract and first paragraph of the introduction gives the impression that this review was going to have a more industrial focus than what is currently written. For example, how widely is HDX-MS used in vaccine development? Only a specific example is given in section 3.3, leaving the reader to wonder if this a one-off example in vaccine development. More broadly, how does each subsection in section 3 relate to clinical development of a therapeutic target? How broadly is HDS-MS used in this context? If the intention was to keep this review more academically focused, then the abstract and first paragraph of the introduction need to be rewritten. Further, there are already reviews in the literature for most of the subsections in section 3 – and in some cases, multiple reviews. Please reference these reviews, so that the interested reader can easily find more information

  • The authors make an inaccurate statement on back exchange. On Page 5, lines 162 – 165, they state,

“Since the H/D exchange reaction is more efficiently base-catalyzed than acid catalyzed, lowering the pH to 2.5 effectively ‘freezes’ the label, reflecting the structural and dynamic characteristics of the protein under ‘native-like’ conditions (prior to the pH drop). This also prevents back-exchange of deuterium with hydrogen in all subsequent steps.”

This statement is factually incorrect. Back exchange rates are not frozen at low pH; they are only reduced by 1000x to 50,000x, depending on the experimental set-up. Practically, 20 – 30 % of the signal is “washed out” from back exchange. For applications of epitope mapping, the issue of back-exchange is less pressing. However, back-exchange has a major impact in later stages of clinical development. In the first reference I give below[1], the HDX-MS  interlab study across 15 labs found the back exchange was 27% averaged across two peptides at 6 immersion times. In the second reference [2], “Back-exchange levels are ideally reported on a per-peptide basis but may be reported as the average percentile loss of deuterium of all peptides analyzed with an indication of the range of values observed, for example, 40% (ranging from 10% to 55%). In a well-conducted conventional state-of-the-art bottom-up HDX-MS workflow, only very few peptides should exhibit back-exchange values above 50%."

[1]          J. W. Hudgens et al., Interlaboratory Comparison of Hydrogen-Deuterium Exchange Mass Spectrometry Measurements of the Fab fragment of NISTmAb. Analytical Chemistry 10.1021/acs.analchem.9b01100 (2019).

[2]          G. R. Masson et al., Recommendations for performing, interpreting and reporting hydrogen deuterium exchange mass spectrometry (HDX-MS) experiments. Nat. Methods 16, 595-602 (2019).

The authors need to include a discussion of back-exchange, including:

- how the problem is mitigated

- how the issue impacts the statistics and how it may impact data interpretation

- why the issue is a little less relevant for epitope mapping

- why, however, it is a very important issue for clinical development for any given protein therapeutic. I am not suggesting that the authors greatly expand the scope by discussing the impact of clinical development in detail, as a thorough discussion would greatly lengthen this section. However, this issue needs to be acknowledged.

  • In Figure 4, we are led to believe that disulfide-linked proteins will be discussed. While many of the proteins subsequently discussed do have disulfide bonds, a section needs to be added on the pros and cons of how these are treated. Are the proteins looked at in an unreduced state? Are the bonds reduced? Why or why not? This has particular applicability to monoclonal antibodies, one of the major drug platforms in the pharmaceutical industry.

  • In section 3.6, please give some indication of how widely HDX-MS is used for membrane proteins. Also, please discuss the impact of the membrane on deuterium uptake? Does this prevent observation of conformational fluctuations? Sample handling Is also a big deal but is only mentioned in passing. Please give more details. This topic in particular has multiple reviews already written about it.

  • For section 3.7, the one example given, a deglycosylated versus a native mAb, represents a major change to the molecule. What is the sensitivity of HDX-MS to more subtle changes to the glycans through glycan remodeling studies? For example, can HDX-MS pick up conformational fluctuations between various glycoforms (e.g., G2F and G0F, etc.)? This is another area that has great relevance to the pharmaceutical industry.

  • What are the challenges that still remain for HDX-MS? This includes protein modalities. What challenges are there for transfer from academia and implementation of wide spread us in the pharmaceutical industry? While this last question could be a full review, the implications of the Hudgens article [1] shows that harmonization for translation of HDX-MS into a pharmaceutical context is a challenge.

  • One general comment: the fonts in several of the figures are too small. These include figures 7, 8, and 11.

Author Response

We would like to thank the reviewers for their careful and constructive review of the manuscript. Detailed responses to reviewer comments and accompanying changes are provided below (changes are noted in italic). We hope that you will find the accompanying revised version of the manuscript suitable for publication in Biomedicines.

Reviewer 3:

- The last sentence of the abstract and first paragraph of the introduction gives the impression that this review was going to have a more industrial focus than what is currently written. For example, how widely is HDX-MS used in vaccine development? Only a specific example is given in section 3.3, leaving the reader to wonder if this a one-off example in vaccine development. More broadly, how does each subsection in section 3 relate to clinical development of a therapeutic target? How broadly is HDS-MS used in this context? If the intention was to keep this review more academically focused, then the abstract and first paragraph of the introduction need to be rewritten. Further, there are already reviews in the literature for most of the subsections in section 3 – and in some cases, multiple reviews. Please reference these reviews, so that the interested reader can easily find more information

Additional references have been added to Section 3.3 to highlight the use of HDX-MS for vaccine and pharmaceuticals development. We highlight in particular ref. [3], which is a review by our group on precisely this topic. While it is true that most of the examples in the current review are academically focused, we attempt to clarify that understanding how proteins behave - for example when binding to ligands or in a pathogenic state - has direct links to therapeutic applications and drug development. We have added references to other recent reviews for the reader’s interest within the subsections.

- The authors make an inaccurate statement on back exchange. On Page 5, lines 162 – 165, they state,

“Since the H/D exchange reaction is more efficiently base-catalyzed than acid catalyzed, lowering the pH to 2.5 effectively ‘freezes’ the label, reflecting the structural and dynamic characteristics of the protein under ‘native-like’ conditions (prior to the pH drop). This also prevents back-exchange of deuterium with hydrogen in all subsequent steps.”

This statement is factually incorrect. Back exchange rates are not frozen at low pH; they are only reduced by 1000x to 50,000x, depending on the experimental set-up. Practically, 20 – 30 % of the signal is “washed out” from back exchange. For applications of epitope mapping, the issue of back-exchange is less pressing. However, back-exchange has a major impact in later stages of clinical development. In the first reference I give below[1], the HDX-MS  interlab study across 15 labs found the back exchange was 27% averaged across two peptides at 6 immersion times. In the second reference [2], “Back-exchange levels are ideally reported on a per-peptide basis but may be reported as the average percentile loss of deuterium of all peptides analyzed with an indication of the range of values observed, for example, 40% (ranging from 10% to 55%). In a well-conducted conventional state-of-the-art bottom-up HDX-MS workflow, only very few peptides should exhibit back-exchange values above 50%."

[1]          J. W. Hudgens et al., Interlaboratory Comparison of Hydrogen-Deuterium Exchange Mass Spectrometry Measurements of the Fab fragment of NISTmAb. Analytical Chemistry 10.1021/acs.analchem.9b01100 (2019).

[2]          G. R. Masson et al., Recommendations for performing, interpreting and reporting hydrogen deuterium exchange mass spectrometry (HDX-MS) experiments. Nat. Methods 16, 595-602 (2019).

The authors need to include a discussion of back-exchange, including:

- how the problem is mitigated

- how the issue impacts the statistics and how it may impact data interpretation

- why the issue is a little less relevant for epitope mapping

- why, however, it is a very important issue for clinical development for any given protein therapeutic. I am not suggesting that the authors greatly expand the scope by discussing the impact of clinical development in detail, as a thorough discussion would greatly lengthen this section. However, this issue needs to be acknowledged.

We agree that the word ‘freeze’ is not appropriate here, and gives the wrong impression. We have modified the sentence to reflect that back-exchange is not eliminated but occurs at a greatly reduced rate. We have also expanded on how the back-exchange issue is handled and cited the two suggested papers for a more in-depth discussion of the matter.

- In Figure 4, we are led to believe that disulfide-linked proteins will be discussed. While many of the proteins subsequently discussed do have disulfide bonds, a section needs to be added on the pros and cons of how these are treated. Are the proteins looked at in an unreduced state? Are the bonds reduced? Why or why not? This has particular applicability to monoclonal antibodies, one of the major drug platforms in the pharmaceutical industry.

When studying disulfide-linked proteins such as monoclonal antibodies, TCEP is added during the quenching step. We have stated this in the introduction and repeated it in Section 3.7, as the monoclonal antibody used in this study was treated with TCEP. We have also added a reference in the introduction which expands on the topic of how to handle disulfide-linked proteins.

- In section 3.6, please give some indication of how widely HDX-MS is used for membrane proteins. Also, please discuss the impact of the membrane on deuterium uptake? Does this prevent observation of conformational fluctuations? Sample handling Is also a big deal but is only mentioned in passing. Please give more details. This topic in particular has multiple reviews already written about it.

We have expanded on experimental details that must be taken in consideration when handling membrane proteins, as well as cited additional recent reviews that go into greater detail on this topic.

- For section 3.7, the one example given, a deglycosylated versus a native mAb, represents a major change to the molecule. What is the sensitivity of HDX-MS to more subtle changes to the glycans through glycan remodeling studies? For example, can HDX-MS pick up conformational fluctuations between various glycoforms (e.g., G2F and G0F, etc.)? This is another area that has great relevance to the pharmaceutical industry.

We have included another recent example that highlights the use of HDX-MS in detecting subtle changes between different glycosylated forms of an isomerase.

- What are the challenges that still remain for HDX-MS? This includes protein modalities. What challenges are there for transfer from academia and implementation of wide spread us in the pharmaceutical industry? While this last question could be a full review, the implications of the Hudgens article [1] shows that harmonization for translation of HDX-MS into a pharmaceutical context is a challenge.

We have included a section discussing about the challenges of HDX-MS as part of the conclusions section.

- One general comment: the fonts in several of the figures are too small. These include figures 7, 8, and 11.

We have enlarged the text of these figures where possible. High-resolution figures can also be included in the electronic version of the manuscript.

Round 2

Reviewer 3 Report

The authors have addressed the majority of my comments satisfactory. The manuscript reads much better now. As such, I have given it a higher rating. It will now be a useful review for the community at large.

One issue: Section 3.3 remains unchanged from V1, even though the authors specifically state that they added references to highlight pharmaceutical and vaccine development. I am guessing the stated changes got lost at some point during the revision process. Please add back in the requested changes for section 3.3.

Author Response

Comment: Section 3.3 remains unchanged from V1, even though the authors specifically state that they added references to highlight pharmaceutical and vaccine development. I am guessing the stated changes got lost at some point during the revision process. Please add back in the requested changes for section 3.3.

Response: We thank reviewer 3 for catching this! We have slightly modified the first sentence of Section 3.3 to make mention of broader applications for HDX-MS in industry before citing [3] and a number of other industry-focused reviews.